# Differential Evolution Particle Swarm Optimization for Phase-Sensitivity Enhancement of Surface Plasmon Resonance Gas Sensor Based on MXene and Blue Phosphorene/Transition Metal Dichalcogenide Hybrid Structure

**DOI:** 10.3390/s23208401

**Published:** 2023-10-12

**Authors:** Chong Yue, Yueqing Ding, Lei Tao, Sen Zhou, Yongcai Guo

**Affiliations:** 1College of Optoelectronic Engineering, Chongqing University, Chongqing 400030, China; ycguo@cqu.edu.cn; 2Chongqing Academy of Metrology and Quality Inspection, Chongqing 401123, China; yueqing_ding@163.com (Y.D.); taol_cqjz@163.com (L.T.); cquzhousen@163.com (S.Z.)

**Keywords:** surface plasmon resonance, gas sensor, phase sensitivity, differential evolution particle swarm optimization, MXene, BlueP/TMDCs

## Abstract

A differential evolution particle swarm optimization (DEPSO) is presented for the design of a high-phase-sensitivity surface plasmon resonance (SPR) gas sensor. The gas sensor is based on a bilayer metal film with a hybrid structure of blue phosphorene (BlueP)/transition metal dichalcogenides (TMDCs) and MXene. Initially, a Ag-BlueP/TMDCs-Ag-MXene heterostructure is designed, and its performance is compared with that of the conventional layer-by-layer method and particle swarm optimization (PSO). The results indicate that optimizing the thickness of the layers in the gas sensor promotes phase sensitivity. Specifically, the phase sensitivity of the DEPSO is significantly higher than that of the PSO and the conventional method, while maintaining a lower reflectivity. The maximum phase sensitivity achieved is 1.866 × 10^6^ deg/RIU with three layers of BlueP/WS_2_ and a monolayer of MXene. The distribution of the electric field is also illustrated, demonstrating that the optimized configuration allows for better detection of various gases. Due to its highly sensitive characteristics, the proposed design method based on the DEPSO can be applied to SPR gas sensors for environmental monitoring.

## 1. Introduction

Surface plasmons (SPs) comprise a special mode of electromagnetic field that exists at the dielectric–metal interface [1,2]. As the resonance absorption peak, resonance angle, and resonance wavelength generated when the surface plasmon resonance (SPR) occurs are closely related to the variation in the refractive index in the sensing medium, sensors find widespread applications in fields such as cell analysis, metamaterial absorbers, and gas detection [3,4,5,6,7,8]. The utilization of SPR in gas detection has garnered significant attention due to its label-free and real-time characteristics [9,10]. However, most SPR sensors rely on measuring the intensity change in the reflected beam, which is susceptible to environmental influences and exhibits a low sensitivity to changes in the refractive index of small molecules [11,12].

In order to promote the sensitivity and resolution of SPR sensors, phase-sensitivity configuration has been proposed in several studies in the literature [13,14,15], based on the fact that the phase of the TM-polarized component of the incident light wave undergoes drastic changes, while the phase of the TE-polarized component remains essentially unchanged [16]. The conventional phase-modulated SPR sensor, which employs the Kretschmann prism coupler, typically utilizes monolayer gold (Au) [17], silver (Ag) [18], aluminum (Al) [19], or copper (Cu) [20] film as the plasmonic metal. Among these metallic materials, Ag is widely favored due to its strong resonance peak resulting from the characteristics of the bulk plasma frequency and low D-electron band [21,22]. However, the inherent absorption of the metal layer broadens the response, and the Ag film is vulnerable to oxidation at ambient temperatures, which negatively affects the phase-sensitivity of the SPR sensor. To mitigate these issues, researchers have proposed the use of two-dimensional (2D) materials. These materials offer a high photoelectric performance, promoting light absorption and providing an improved biological compatibility.

Blue phosphene (BlueP) is a promising 2D material produced by Zhu and Tomanek, following the experimental discovery of black phosphorus [23]. The lattice arrangement of phosphorus atoms in BlueP is located in the lower curved honeycomb of its lattice [24]. BlueP demonstrates thermal stability comparable to black phosphorus, and also features a band gap of 2ev, making it highly suitable for application in SPR sensors [25]. Additionally, due to the hexagonal crystal structure shared by the BlueP and TMDC monolayers, it is possible to construct a van der Waals (vdw) hybrid structure of BlueP/TMDCs [26]. This vdw heterostructure exhibits novel optoelectronic characteristics previously unseen in 2D materials. BlueP-based SPR phase sensors have been utilized in practical biosensing due to their superior sensitivity compared to traditional monolayer biosensors [27,28,29]. Liao successfully developed an SPR sensor using ITO and BlueP/MoS_2_, achieving a maximum phase sensitivity of ~3.600 × 10^6^ deg/RIU [30]. Furthermore, Li proposed an SPR biosensor based on BlueP–graphene, which achieved the highest phase sensitivity of 1.473 × 10^5^ deg/RIU [31].

MXene consists of metal carbide and metal nitride materials with a 2D layered structure [32]. It is exfoliated via the MAX phase and has a universal chemical formula M_n+1_AX_n_ (n = 1, 2, or 3), where M represents early transition metals, A usually represents chemical elements of the third or fourth main group, and X represents nitrogen or carbon [33,34,35]. Due to its large surface-to-volume ratio and metal level conductivity, MXene significantly enhances the detection limit of small molecules for SPR sensors modified with MXene. However, the sensors discussed in the aforementioned literature employ a conventional method (CM) based on layer-by-layer optimization. As the number of SPR sensor layers increases, this method becomes time-consuming and makes it challenging to determine the optimal thickness of all the layers simultaneously.

An intelligent optimization algorithm is developed to design a multi-layer SPR sensor for high sensitivity and resolution. The structure and performance of SPR sensors are optimized using the particle swarm optimization (PSO) algorithm. Sun proposed a SPR biosensor structure with the PSO algorithm and analyzed its sensing performance under four different modulation modes (wavelength, angle, phase, intensity) [36]. The results demonstrate that the optimization structure based on the PSO algorithm outperforms the experimental structure. Additionally, Amoosoltani used the PSO algorithm to optimize the thickness of the metal layer in an SPR gas sensor [37]. The simulation results show a significant improvement in the Q-factor and FWHM using this method. However, the PSO algorithm is prone to parameter selection and multi-objective parameter optimization, leading to a faster convergence in the early stage and slower convergence in the later stage, and a higher likelihood of being trapped in the local optima [38]. To address these limitations, the differential evolution (DE) algorithm introduces mutation and crossover concepts into the particle position update, enabling the algorithm to escape local optima and effectively handle a large number of design parameters.

The DEPSO algorithm is used in this article to design a high-phase-sensitivity SPR gas sensor based on the Ag-BlueP/TMDCs-Ag-MXene hybrid structure for gas detection. The DEPSO algorithm uses an objective function with a restraint condition of minimum reflectivity. By optimizing the layer thickness in the Ag-BlueP/TMDCs-Ag-MXene modified structure, the phase sensitivity of the SPR gas sensor can be increased while maintaining a low reflectivity, making it more suitable for gas detection. The performance of the optimized structure is then evaluated via detecting different gases. Finally, the ultrasensitive properties of the optimized structure are further analyzed via illustrating the electric field distribution.

## 2. Theoretical Modeling and Design Consideration

The schematic diagram of the phase-sensitivity SPR gas-detection setup is shown in Figure 1. The proposed SPR gas sensor consists of a BK7 prism ($n_{BK7}=1.5151$); a Ag film with RI is $n_{Ag}=0.0803+4.2347i$ based on the Drude–Lorentz model as the plasmonic metal. Table 1 shows the RIs and monolayer thicknesses of the BlueP/TMDCs and MXene at the wavelength of 633 nm. The sensing medium is air, and its RI is 1. To provide a basis for the numerical simulation method, Figure 1 shows the feasibility of the proposed method in the experiment. Firstly, the He-Ne laser passes through a laser beam expander and polarizer to obtain 45° linear polarization for the P-wave and S-wave. Then, the light is incident upon the SPR gas sensor at angle θ, which is controlled by the 3-axis rotation stage. At last, it enters the Mach–Zehnder interference to obtain the interference patten and calculate the actual phase sensitivity of the SPR gas sensor.

For the sake of validating the performance of the sensor, the transfer matrix method (TMM) is used to compute the total reflection coefficient *r* of p-polarized and s-polarized light, as follows:(1)$$r=\frac{\left(M_{11}+M_{12}q_{N})q_{0}-(M_{21}+M_{22}q_{N}\right)}{\left(M_{11}+M_{12}q_{N})q_{0}+(M_{21}+M_{22}q_{N}\right)}$$

The characteristic matrix of the N-layer structure is given by:(2)$$\begin{array}{ll} M=\prod_{m=1}^{N}M_{m} & =\left[\begin{array}{cc} M_{11} & M_{12}\\ M_{21} & M_{22} \end{array}\right]\\ & =\left[\begin{array}{cc} \cos\beta_{m} & -\frac{i}{q_{m}}\sin\beta_{m}\\ -q_{m}\sin\beta_{m} & \cos\beta_{m} \end{array}\right] \end{array}$$
(3)$$\left\{\begin{array}{l} q_{k}=\frac{{\left(\varepsilon_{k}-n_{0}^{2}\sin^{2}\theta_{0}\right)}^{\frac{1}{2}}}{\varepsilon_{k}}\;\;P\text{-}wave(TM)\\ q_{k}={\left(\varepsilon_{k}-n_{0}^{2}\sin^{2}\theta_{0}\right)}^{\frac{1}{2}}\;\;S\text{-}wave(TE) \end{array}\right.$$
(4)$$\beta_{m}=\frac{2\pi d_{m}}{\lambda }{\left(\varepsilon_{m}-n_{0}^{2}\sin^{2}\theta_{0}\right)}^{\frac{1}{2}}$$

In these equations, λ represent the wavelength of the incident light, and $q_{k}$ is the optical admittances. $d_{m}$ and $\varepsilon_{m}$ represent the thickness and dielectric constant of each film layer, respectively. From the above equation, the reflectance *R_p_* is $R_{p}={\left|r_{p}\right|}^{2}$, and the phase of the p-polarized and s-polarized light can be expressed as:(5)$$\left\{\begin{array}{l} \phi_{p}=\arg(r_{p})\\ \phi_{s}=\arg(r_{s}) \end{array}\right.$$

Furthermore, the phase difference ($\phi_{d}$) between the p-polarized ($\phi_{p}$) and s-polarized ($\phi_{s}$) light can be described as:(6)$$\phi_{d}=\left|\phi_{p}-\phi_{s}\right|$$

Therefore, the phase sensitivity of the sensor is given by:(7)$$S=\frac{\Delta \phi_{d}}{\Delta n_{bio}}$$
where $\Delta \phi_{d}$ is the differential phase corresponding to changes in the refractive index of gas.

In this work, a high-phase-sensitivity SPR sensor based on a Ag-BlueP/TMDCs-Ag-MXene hybrid structure is designed and numerically investigated using MATLAB software (R2021a), based on differential evolution particle swarm optimization and the transfer matrix method with the Fresnel equation for the detection of various gases.

## 3. Differential Evolution Particle Swarm Optimization

The particle swarm optimization (PSO) is a group-based intelligence optimization algorithm that simulates the foraging behavior of birds. It achieves this by imagining each bird as a particle and representing each possible solution as a particle in the population [39]. Each particle has its own velocity and position. The updating of the particle position and velocity is mainly achieved via comparing itself with the surrounding particles and the current optimal value of the population. Due to these characteristics, it can be seen that the PSO is suitable for optimizing the solution of multi-dimensional problems. However, the PSO has the problem of a fast convergence speed in the early stage. Additionally, the information exchange between each particle is unidirectional, which causes the population to lack diversity in the later stages of the algorithm and easily fall into the local optimum.

To address this issue, this paper designs a differential evolution particle swarm optimization algorithm. The idea is to introduce mutation and crossover strategies from the differential evolution algorithm into the process of particle updates. This integration aims to maintain the diversity of the particle swarm in the later stage of the search, thereby improving the global optimization performance of the algorithm. 

The specific steps of designing the phase-sensitivity SPR gas sensor based on the DEPSO are as follows. At first, the population positions and velocities need to be randomly initialized, and then the positions and velocities of the particles are updated after a comparison. Afterwards, we perform mutation operations on the updated population to generate temporary intermediate individuals:(8)$$\omega_{i,G+1}=x_{best,G}+F\cdot (x_{r_{1},G}-x_{r_{2},G}+x_{r_{3},G}-x_{r_{4},G})$$
where the randomly selected serial numbers *r*_1_, *r*_2_, *r*_3_, and *r*_4_ are different from each other, and *F* denotes the scaling factor, which determines the degree of variation.

Similar to the crossover operator in genetic algorithms, in order to enhance the population diversity, the crossover operation in the differential algorithm can be represented as:(9)$$\mu_{ij,G+1}=\left\{\begin{array}{ll} \omega_{ij,G+1}, & if\;randb(j)\leq CR\;or\;j=j_{rand}\\ x_{ij,G+1}, & otherwise \end{array}\right.$$
where CR represents the crossover probability, and *j_rand_* is a random integer between [1, 2, …, D].

Finally, new populations are generated through the above mutation and crossover operation, and the new populations are evaluated and the global optimal solution is updated until the termination condition is satisfied. The pseudocode of the DEPSO can be described as below (Algorithm 1):
**Algorithm 1: DEPSO**   Initialize:(1) Population N, dimension D, iteration T, Scaling factor F, Leaning factor C, hybrid probability(2) Randomly initialize particle position x_pso, velcolity v, mutation operator w, selection operator u, P_i_ and P_g_ of particles(3) Cycle(4) **For** i = 1:N(5)    **For** j = 1:D(6)      vij(t+1)=vij(t)+c1r1(pij(t)−x_psoij(t))               +c2r2(pgj(t)−x_psoij(t))x_psoij(t+1)=x_psoij(t)+vij(t+1)       %Update the velocity and position of the particle(7)       **If** func(x_psoij) > func(pij) **then**
pij = x_psoij(8)       **End If** func(x_psoij) > func(pgj) **then**
pgj=x_psoij(9)       **End**       % Mutation and Crossover operation(10)      F=F0⋅2e1−TT+1−t(11)      ωi=xbest+F(xr1−xr2+xr3−xr4)(12)      uij= Crossover (x_psoi,ωi)(13)        **If** func(ui) > func(x_psoi) **then**
xi = ui(14)        **else**
xi = x_psoi(15)        **End**       % Update globe values(16)        **If** max(func(pg)) > max(func(xi)) then(17)       [fmax,r] = max(func(pg)); bestx = x_pso(r,:);()(18)        **else** [fmax,r] = max(func(xi)); bestx = xi(r,:);(19)        **End**(20)    **End**(21) **End**


The DEPSO is used to optimize the thickness of the proposed structure to simultaneously obtain a high phase sensitivity and low reflectivity at the resonance angle. Therefore, the objective function is defined as:(10)$$OF=\left\{\begin{array}{l} S,R_{\min}<0.01\\ 0,\;others \end{array}\right.$$
where *S* is the phase sensitivity, and *R*_min_ represents the minimum reflectivity. The aim is to find the maximum value of the objective function in the search region. Apart from that, the process of finding the maximum phase sensitivity is also the process required to minimize the value of *R*_min_; if the *R*_min_ is larger than 0.01, then the solution will be discarded.

## 4. Results and Discussion

In order to verify the effectiveness of the proposed method, the conventional method and particle swarm optimization (PSO) are used to optimize and verify the same heterostructure. First of all, in the conventional method, the thickness of the first layer(*d*_1_) and the third layer(*d*_3_) of the Ag film are randomly set to 20 nm, the RI of the gas is changed to 0.0001, and the phase sensitivity with respect to different SPR sensor structures at the same change in RI are obtained. As shown in Figure 2, the sensor structures (I~IV) represent Ag-BlueP/TMDCs (BlueP/MoS_2_, BlueP/MoSe_2_, BlueP/WS_2_, BlueP/WSe_2_)-Ag-MXene hybrid structure, N stands for BlueP/TMDCs, and L is for the MXene layer. It can be seen from Figure 2 that, without the layer of BlueP/TMDCs and MXene, the sensor structure has the lowest phase sensitivity, with 4.875 × 10^4^ deg/RIU. Then, by adding monolayer BlueP/TMDCs or MXene, the phase sensitivity significantly increases compared to the sensor structure that only has the coated Ag film. Moreover, the sensor structures (I~IV) show the highest phase sensitivity, with 1.582 × 10^5^ deg/RIU, 1.611 × 10^5^ deg/RIU, 1.514 × 10^5^ deg/RIU, 1.542 × 10^5^ deg/RIU when both the monolayer of BlueP/TMDCs and MXene are added. Through a comparison of the corresponding data, it can be concluded that the phase sensitivity of the hybrid structure (I~IV) *N* = 1&*L* = 1 is more than three times that of *N* = 0&*L* = 0, whereas the improvement in phase sensitivity for *N* = 1&*L* = 0 over *N* = 0&*L* = 0 is more than 2%, and the phase sensitivity of structure *N* = 0&*L* = 1 is less than three times that of *N* = 0&*L* = 0. Therefore, the proposed SPR gas sensor exhibits a greater enhancement compared to the sensor structure that only has a monolayer of BlueP/TMDCs, MXene, or Ag film.

In addition, to better illustrate the improvement in the phase sensitivity of the SPR sensor through the addition of the BlueP/TMDCs and MXene layer, the influence of the number of layers of BlueP/TMDCs and MXene on the phase sensitivity is shown in Figure 3. It is clear from the figure that the phase sensitivity is related to the number of BlueP/TMDCs and MXene layers, with the increase in layers, the phase sensitivity is not a monotone function. The highest phase sensitivities are 5.983 × 10^5^ deg/RIU, 5.767 × 10^5^ deg/RIU, 9.155 × 10^5^ deg/RIU, 9.492 × 10^5^ deg/RIU for the five-layer Blue/MoS_2_, four-layer Blue/MoSe_2_, seven-layer Blue/WS_2_, and eight-layer Blue/WSe_2_ with a monolayer of MXene, respectively. 

It can be concluded from the above analysis that the sensor structure with the BlueP/TMDCS and MXene layer is feasible in improving the phase sensitivity. However, the number of BlueP/TMDCs and MXene layers cannot be arbitrarily determined. Therefore, it is necessary to use an algorithm to simultaneously optimize the thickness of each layer of the SPR sensor.

A Ag-BlueP/TMDCs-Ag-MXene hybrid-structure-based SPR gas sensor is designed via the PSO and the DEPSO to verify the effectiveness of the method. In Equations (1)–(8), it is evident that the value of the objective function is related to the thickness of Ag *d*_1_ and *d*_2_, the thickness of BlueP/TMDCs *N*_1_–*N*_4_, and the number of layers in MXene *L*_1_. Thus, there are seven design parameters, which are as follows: *x* = [*d*_1_, *d*_2_, *N*_3_, *N*_4_, *N*_5_, *N*_6_, *L*_7_]*^T^* = [*x*_1_, *x*_2_, *x*_3_, *x*_4_, *x*_5_, *x*_6_, *x*_7_]*^T^*. The search range for each variable is set as $0<d_{1},d_{2}<50\;\mathrm{nm}$, $0<N_{1}\ldots N_{4},L_{1}\leq 10$ layers. At 633 nm, silver film thicknesses (*d*_1_, *d*_2_) less than 50 nm usually have a good sensing performance [16,18] and, for $N_{1}\ldots N_{4},L_{1}$ > 10 layers, the resonant dip vanishes. Before invoking the algorithm, the initialization variables are listed in Table 2, all of which are set based on experience to enable the algorithm to perform optimally [40]. 

After 100 iterations, the change in the optimized layer thickness of the Ag-BlueP/TMDCs-Ag-MXene structure-based SPR sensor via the PSO is shown in Figure 4. The corresponding structural parameters, including the minimum reflectivity at the resonance angle and the phase sensitivity, are presented in Table 3. From Figure 4 and the relevant data in Table 3, it can be seen that as the number of iterations increases, the thickness of the optimized SPR biosensor oscillates first and then converges to a stable value. For Figure 4a and Table 2, the optimized Ag-BlueP/MoS_2_-Ag-MXene heterostructure reaches the highest phase sensitivity of 1.824 × 10^6^ deg/RIU, the minimum reflectivity of 2.360 × 10^−4^, when the thickness of the first and third Ag are 19.064 nm and 22.244 nm, and the BlueP/MoS_2_ and MXene are bilayer and monolayer, respectively. For the Ag-BlueP/MoSe_2_-Ag-MXene heterostructure, when the BlueP/MoSe_2_ and MXene are both monolayer, the thickness of the first and third Ag layer are 16.310 nm and 25.367 nm, the highest phase sensitivity is 1.808 × 10^6^ deg/RIU, and the minimum reflectivity is 5.289 × 10^−5^. In Figure 4c, the optimized Ag-BlueP/WS_2_-Ag-MXene hybrid structure achieves the highest phase sensitivity of 1.821 × 10^6^ deg/RIU, the minimum reflectivity of 1.271 × 10^−4^, when the thickness of first and third Ag layer are 16.310 nm and 25.367 nm, and the layer of both the BlueP/WS_2_ and graphene is a monolayer. Finally, the SPR sensor of the Ag-BlueP/WSe_2_-Ag-MXene heterostructure is optimized when the BlueP/WSe_2_ and MXene are monolayer, the thickness of first and third Ag layer are 21.502 nm and 20.142 nm, the highest phase sensitivity can achieve 1.816 × 10^6^ deg/RIU, and the minimum reflectivity is 1.395 × 10^−4^. The above results demonstrate that, by optimizing the thickness of each layer, the phase-sensitivity to RI change can be promoted in the SPR gas sensor.

Figure 5 presents the schematic of the implementation of the DEPSO algorithm in conjunction with TMM. Initially, a population of 100 configurations is generated, and the phase difference and minimum reflectivity for each configuration are computed. Subsequently, the configurations are sorted based on their phase sensitivity, given that the minimum reflectance is less than 0.01. Following this, the position and velocity of the particles are updated, and mutation and crossover operations are performed to generate the offspring population. Furthermore, TMM calculates the phase sensitivity of the offspring population and compares the OF value with the current optimal solution to determine the configuration to be transferred to the next generation. Ultimately, the entire process is repeated until the termination condition is achieved, leading to the identification of the best solution.

Figure 6 and Table 4 gives the change in the optimized layer thickness of the Ag-BlueP/TMDCs-Ag-MXene structure-based SPR sensor via the DEPSO. Under the premise of ensuring a minimum reflectivity of less than 0.01, it can be seen that, compared with the PSO and the conventional method, the phase sensitivity based on the DEPSO is greatly improved, and the number of iterations of the algorithm to reach the stable optimal value is significantly reduced. For the BlueP/MoS_2_ structure, for the highest phase sensitivity of 1.833 × 10^6^ deg/RIU, the minimum reflectivity is 1.824 × 10^−4^, when the thickness of the first and third Ag film are 18.401 nm and 22.609 nm, the BlueP/MoS_2_ is three layers, and the MXene is one layer. Subsequently, the optimized Ag-BlueP/MoSe_2_-Ag-MXene structure via the DEPSO is shown as Figure 6b. The highest phase sensitivity is 1.811 × 10^6^ deg/RIU, the minimum reflectivity of 3.597 × 10^−5^, when the thickness of the first and third Ag are 17.668 nm and 22.948 nm, the BlueP/MoSe_2_ is four layers, and the MXene is one layer. Then, when the BlueP/WS_2_ is three layers, the MXene is one layer, and the thickness of the first and third Ag layers are 18.513 nm and 22.712 nm, the maximum phase sensitivity can reach 1.866 × 10^6^ deg/RIU for the Ag-BlueP/WS_2_-Ag-MXene hybrid structure. Finally, for the SPR gas sensor of the Ag-BlueP/WSe_2_-Ag-MXene heterostructure, when the BlueP/WSe_2_ is bilayer and the MXene is monolayer, and the thicknesses of the first and third Ag films are 17.226 nm and 24.236 nm, the maximum phase sensitivity can reach 1.821 × 10^6^ deg/RIU, and the minimum reflectivity is 1.283 × 10^−4^. In comparing Figure 5 with Figure 6, and Table 3 with Table 4, it can be seen that the DEPSO algorithm requires fewer iterations to reach an optimal solution, which shows better convergence characteristics. Moreover, the DEPSO algorithm exhibits a superior global search ability, which enables it to jump out of the local optimal value and more fully utilize the high electron concentration and mobility of BlueP/TMDCs and MXene, thus enhancing the performance of SPR sensors, making them suitable for detecting mixed gases. At the same time, as the number of SPR sensor layers increases, the DEPSO algorithm can find the optimal thickness of all layers simultaneously and save time, which has the advantage of working with a large number of design parameters.

At the same time, the objective function curves of the Ag-BlueP/TMDCs-Ag-MXene hybrid structure of the SPR gas sensor via the PSO and the DEPSO are shown in Figure 7. It is observed that, with increasing iterations, the value of the objective function of the DEPSO increases much more rapidly than the PSO, and the higher value of OF has been stable for about 30 times when the value of the objective function of the PSO has been stable for about 65 times. Based on the above analysis, it can be seen that the DEPSO algorithm not only effectively improves the phase sensitivity of the SPR gas sensor, but also reduces the number of iterations. Therefore, the DEPSO has a high efficiency and accuracy in optimizing the multi-layer sensor structure.

To further illustrate the performance of the optimized sensor, the electric field distribution has been investigated in Figure 8. From (a) to (d), one can see that, after coating the BlueP/TMDCs and MXene onto the surface of the traditional SPR sensor, there is a great increase in the electric field intensity, which means a stronger excitement of SPs. Furthermore, the electric field intensity shows tremendous changes at the MXene–sensing-medium interface when the RI of the air changes from 1 to 1.001, which illustrates that a small change in the sensing medium will result in dramatic variation in the surface wave characteristics. 

In addition, Figure 9 shows that the visible wavelength range has been utilized to study the effect of changes in phase difference on sensitivity. From the figure, it can be observed that the highest phase sensitivity can be obtained around 633 nm but, in other visible regions, the proposed sensing structure in this paper also achieves a high phase sensitivity. This indicates that the proposed sensor structure can be used not only to detect gases at 633 nm, but also to detect gases at different wavelengths.

Moreover, different gases are used to benchmark the adaptivity of the proposed sensor at 633 nm. From Table 5, it can see that the sensor structures optimized in this paper had a high phase sensitivity when detecting different gases, which indicates that the BlueP/TMDCs and MXene are suitable for SPR gas sensor due to their unique characteristics.

## 5. Conclusions

A novel method for designing SPR gas sensors with a high phase sensitivity using bilayer metal film, BlueP/TMDCs, and MXene is proposed in this work. It has been proven that the DEPSO method, with restrained conditions to simultaneously optimize the thickness of each layer of the SPR gas sensor, exhibits a better efficiency and accuracy compared to the conventional and PSO methods. The results indicate that the sensor with the Ag-BlueP/TMDCs-Ag-MXene structure, where BlueP/WS_2_ has three layers and MXene is monolayer, achieves the maximum phase sensitivity of 1.833 × 10^6^ deg/RIU, 1.811 × 10^6^ deg/RIU, 1.866 × 10^6^ deg/RIU, and 1.821 × 10^6^ deg/RIU. Furthermore, the ultrasensitive properties are demonstrated by the distribution of the electric field and the detection of different gases, which support the suitability of the proposed structure for gas-sensing applications.

## Figures and Tables

**Figure 1 sensors-23-08401-f001:**
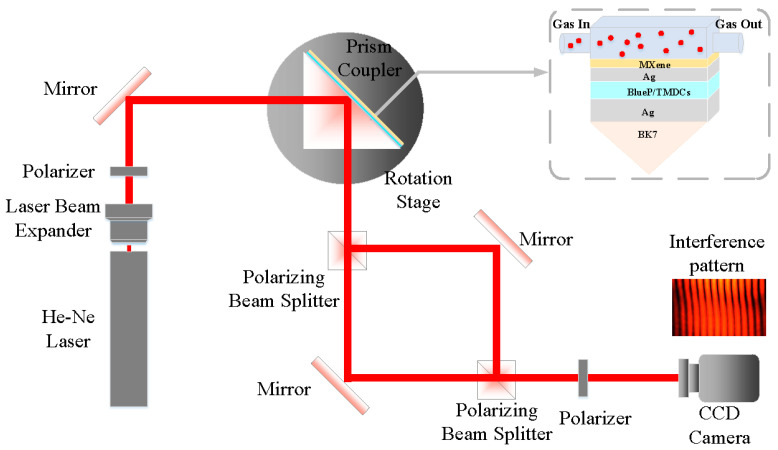
Schematic diagrams of the proposed phase-sensitivity SPR gas-detection setup.

**Figure 2 sensors-23-08401-f002:**
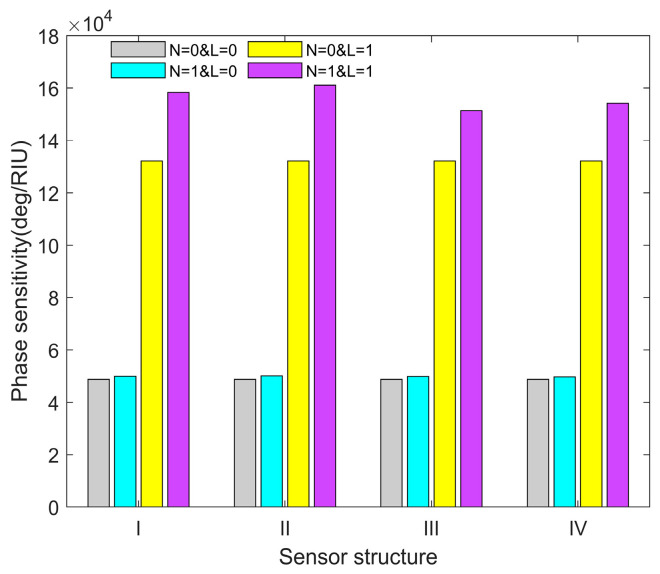
The phase sensitivity with respect to the different SPR sensor structures at the same change in RI.

**Figure 3 sensors-23-08401-f003:**
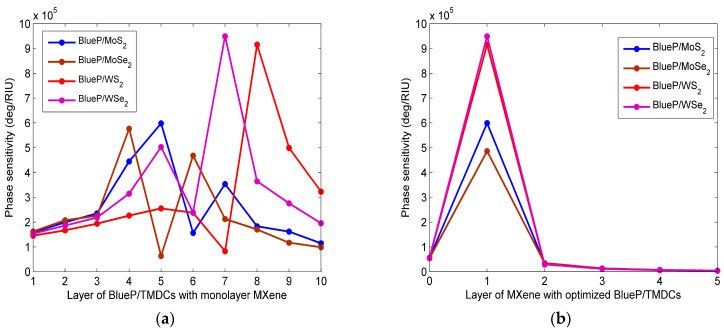
The phase sensitivity with (**a**) the number of BlueP/TMDCs layers under the condition of monolayer Mxene; (**b**) the number of Mxene layers under the condition of optimized BlueP/TMDCs.

**Figure 4 sensors-23-08401-f004:**
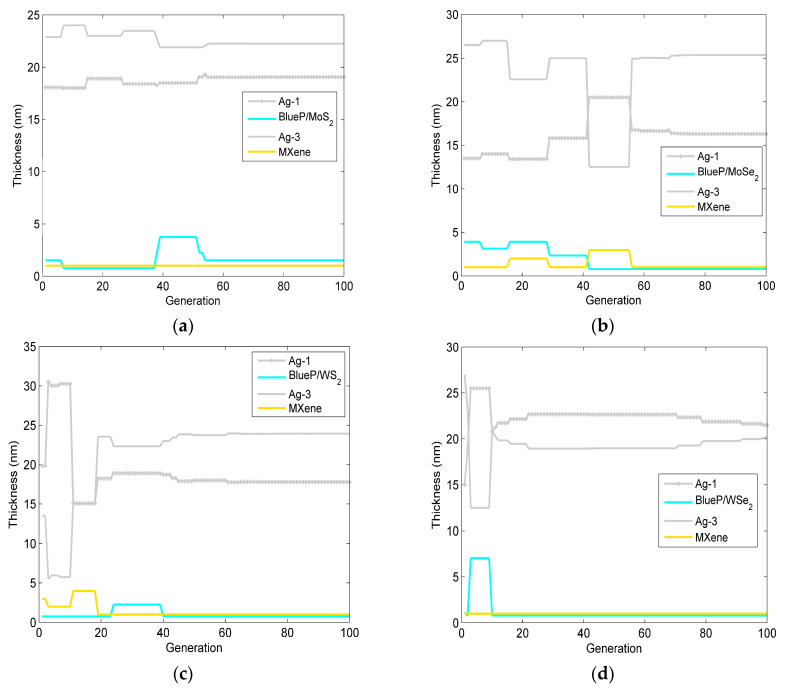
The change in the optimized layer thickness of the designed SPR gas sensor versus the generation of the PSO. (**a**) Ag-BlueP/MoS_2_-Ag-MXene structure; (**b**) Ag-BlueP/MoSe_2_-Ag-MXene structure; (**c**) Ag-BlueP/WS_2_-Ag-MXene structure; (**d**) Ag-BlueP/WSe_2_-Ag-MXene structure.

**Figure 5 sensors-23-08401-f005:**
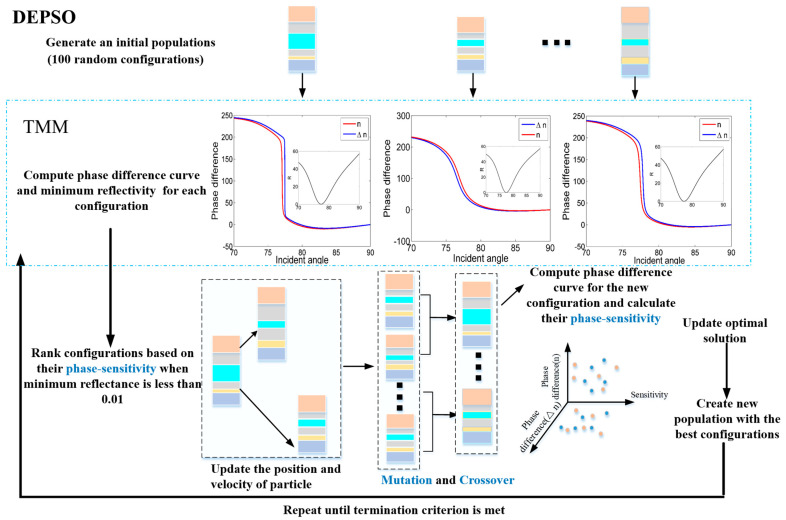
Schematic of the implementation of the DEPSO alongside TMM.

**Figure 6 sensors-23-08401-f006:**
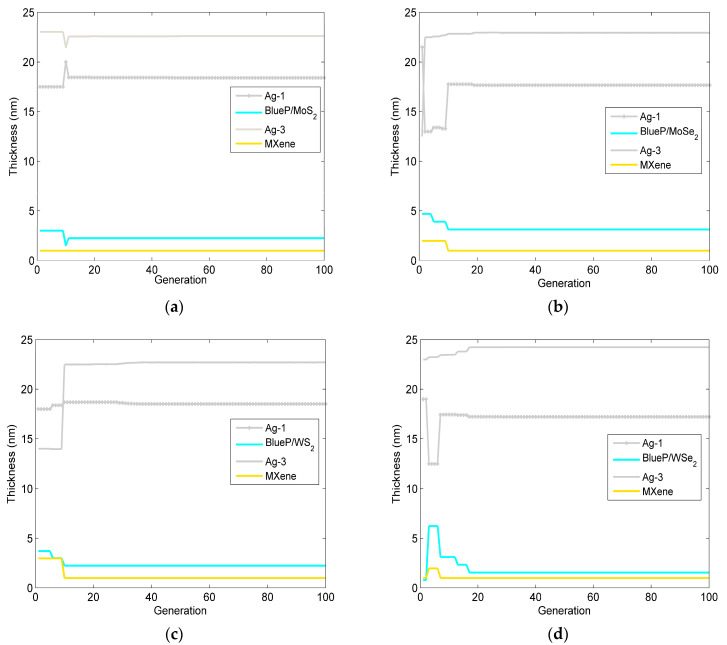
The change in optimized layer thickness of the designed SPR gas sensor versus generation via the DEPSO. (**a**) Ag-BlueP/MoS_2_-Ag-MXene structure; (**b**) Ag-BlueP/MoSe_2_-Ag-MXene structure; (**c**) Ag-BlueP/WS_2_-Ag-MXene structure; (**d**) Ag-BlueP/WSe_2_-Ag-MXene structure.

**Figure 7 sensors-23-08401-f007:**
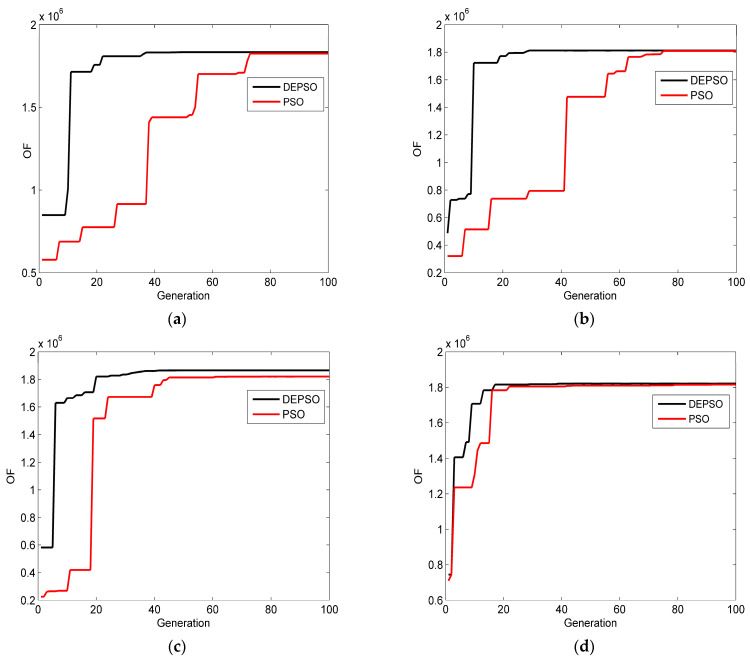
The change in the optimized OF of the designed SPR gas sensor versus the generation of the DEPSO and the PSO. (**a**) Ag-BlueP/MoS_2_-Ag-MXene structure; (**b**) Ag-BlueP/MoSe_2_-Ag-MXene structure; (**c**) Ag-BlueP/WS_2_-Ag-MXene structure; (**d**) Ag-BlueP/WSe_2_-Ag-MXene structure.

**Figure 8 sensors-23-08401-f008:**
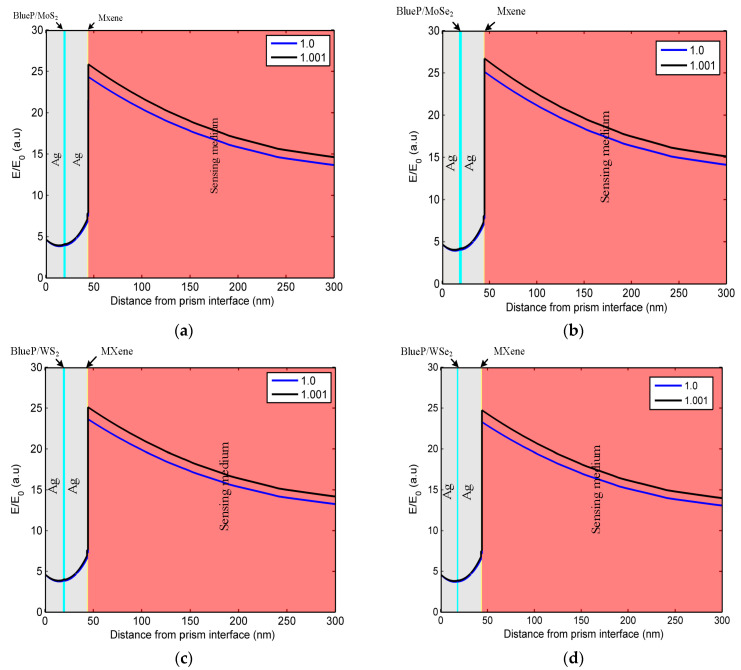
The electric field intensity with the distance from the prism interface for the (**a**) Ag-BlueP/MoS_2_-Ag-MXene structure; (**b**) Ag-BlueP/MoSe_2_-Ag-MXene structure; (**c**) Ag-BlueP/WS_2_-Ag-MXene structure; (**d**) Ag-BlueP/WSe_2_-Ag-MXene structure.

**Figure 9 sensors-23-08401-f009:**
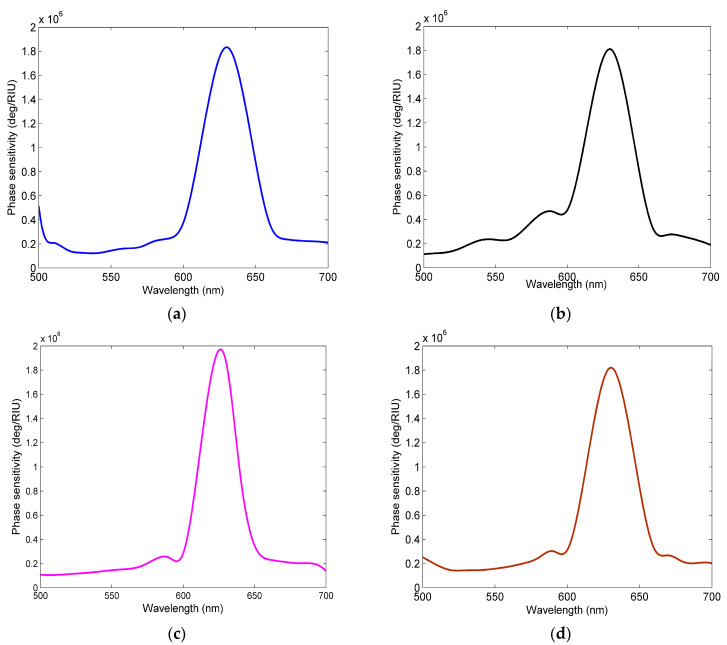
The phase sensitivity with respect to the wavelength for the (**a**) Ag-BlueP/MoS_2_-Ag-MXene structure; (**b**) Ag-BlueP/MoSe_2_-Ag-MXene structure; (**c**) Ag-BlueP/WS_2_-Ag-MXene structure; (**d**) Ag-BlueP/WSe_2_-Ag-MXene structure.

**Table 1 sensors-23-08401-t001:** The monolayer RI and thickness of BlueP/TMDCS and MXene at λ = 633 nm [26,32].

2D Materials	Thickness (nm)	Refractive Index
BlueP/MoS_2_	0.75	2.81 + 0.32i
BlueP/MoSe_2_	0.78	2.77 + 0.35i
BlueP/WS_2_	0.75	2.48 + 0.17i
BlueP/WSe_2_	0.78	2.69 + 0.22i
MXene	0.99	2.38 + 1.33i

**Table 2 sensors-23-08401-t002:** The setting of the parameters for the algorithms.

Parameters	Algorithm	
	PSO	DEPSO
Particle number	100	100
Maximum iterative times	100	100
Acceleration constants c1/c2	2/2	/
Inertia weight coefficient range	[0.6, 0.9]	[0.6, 0.9]
Scale factor F_0_	/	0.1
Crossover probability CR	/	[0.2, 0.9]

**Table 3 sensors-23-08401-t003:** The performance parameters of the optimized Ag-BlueP/TMDCs-Ag-MXene structure via the PSO.

Type of BlueP/TMDCs	First Layer Ag (nm)	Third Layer Ag (nm)	BlueP/TMDCs Layer (N)	Mxene Layer (L)	Minimum Reflectivity	Incident Angle (deg)	Phase Sensitivity (deg/RIU)	Iterations (Times)
BlueP/MoS_2_	19.064	22.244	2	1	2.360 × 10^−4^	43.09	1.824 × 10^6^	58
BlueP/MoSe_2_	16.310	25.367	1	1	5.289 × 10^−5^	43.06	1.808 × 10^6^	67
BlueP/WS_2_	17.806	23.932	1	1	1.271 × 10^−4^	43.06	1.821 × 10^6^	60
BlueP/WSe_2_	21.502	20.142	1	1	1.395 × 10^−4^	43.07	1.816 × 10^6^	93

**Table 4 sensors-23-08401-t004:** The performance parameters of the optimized Ag-BlueP/TMDCs-Ag-MXene structure via the DEPSO.

Type of BlueP/TMDCs	First Layer Ag (nm)	Third Layer Ag (nm)	BlueP/TMDCs Layer (N)	Mxene Layer (L)	Minimum Reflectivity	Incident Angle (deg)	Phase Sensitivity (deg/RIU)	Iterations (Times)
BlueP/MoS_2_	18.401	22.609	3	1	1.824 × 10^−4^	43.11	1.833 × 10^6^	13
BlueP/MoSe_2_	17.668	22.948	4	1	3.597 × 10^−5^	43.13	1.811 × 10^6^	21
BlueP/WS_2_	18.513	22.712	3	1	1.271 × 10^−4^	43.10	1.866 × 10^6^	32
BlueP/WSe_2_	17.226	24.236	2	1	1.283 × 10^−4^	43.08	1.821 × 10^6^	18

**Table 5 sensors-23-08401-t005:** The performance parameters of the Ag-BlueP/WS_2_-Ag-MXene structure for various gases.

Gas	RI	Minimum Reflectivity	Phase Sensitivity (deg/RIU)
Helium He	1.000035 [41]	9.177 × 10^−5^	1.822 × 10^6^
Ethane C_2_H_6_	1.000748 [42]	4.901 × 10^−5^	1.833 × 10^6^
Propane C_3_H_8_	1.00108 [43]	1.022 × 10^−4^	1.836 × 10^6^
Hydrogen H_2_	1.000132 [41]	1.552 × 10^−4^	1.829 × 10^6^

## Data Availability

Not applicable.

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
