# Peer review of "Differential Evolution Particle Swarm Optimization for Phase-Sensitivity Enhancement of Surface Plasmon Resonance Gas Sensor Based on MXene and Blue Phosphorene/Transition Metal Dichalcogenide Hybrid Structure"

_sensors, 2023, doi:10.3390/s23208401_

Round 1

Reviewer 1 Report

In this work, the authors propose a differential evolution particle swarm optimization (DEPSO) design of a high phase sensitive surface plasmon resonance (SPR) gas sensor based on a hybrid structure of blue phosphorene (BlueP)/transition metal disulfide compounds (TMDCs) and MXene bilayer metal films. In virtue of highly sensitive characteristics, the proposed design method based on DEPSO can be applied to SPR gas sensor for environmental monitoring. I believe that publication of the manuscript may be considered only after the following issues have been resolved.

1.    The text information in Figure 5 is not clear enough, and the author needs to make adjustments.

2.    Since this work is a type of sensor, the authors need to provide a sensing simulation spectrum of the device.

3.    The authors need to provide the physical method or software for the design of this sensor.

4.    The introduction can be improved. The articles related to some applications of SPR sensors should be added such as Micromachines 2023, 14(5), 953; Journal of Lightwave Technology, 39(5):1544-1549, 2021; Electronics 2023, 12(12), 2655; IEEE Photonics Journal, 12(3), 4500209, 2020.

5.    The English expression of the whole article needs to be further improved.

 Minor editing of English language required.

Author Response

Comments 1: The text information in Figure 5 is not clear enough, and the author needs to make adjustments.

Response 1: Thank you for pointing this out. I agree with this comment. Therefore, I have adjusted the text information in Figure 5 and added legends for the colors in the curves in the TMM subfigures for better clarity.

Comments 2: Since this work is a type of sensor, the authors need to provide a sensing simulation spectrum of the device.

Response 2: Thank you for pointing this out. As this article focuses on optimizing sensor structure to enhance phase sensitivity, in Figure 9, I provide phase sensitivity with respect to the wavelength in the visible region of the device. And indicates that the proposed sensor structure can be used not only to detect gases at 633nm, but also has high phase sensitivity to detect gases at different wavelengths.

Comments 3:  The authors need to provide the physical method or software for the design of this sensor.

Response 3: Thank you for pointing this out. I added the description of physical method and software for the design of this sensor in last paragraph of section 2.

Comments 4:  The introduction can be improved. The articles related to some applications of SPR sensors should be added such as Micromachines 2023, 14(5), 953; Journal of Lightwave Technology, 39(5):1544-1549, 2021; Electronics 2023, 12(12), 2655; IEEE Photonics Journal, 12(3), 4500209, 2020.

Response 4: Thank you for pointing this out. I have revised the introduction and added the above articles related to some applications of SPR sensors in reference [4-7].

Comments 5:  The English expression of the whole article needs to be further improved.

Response 5: Thank you for pointing this out. I have revised some of the English expressions in the article, but the level is limited, and I hope the professor will criticize and correct me.

Reviewer 2 Report

The authors introduced the Differential Evolution Particle Swarm Optimization (DEPSO) method to determine best parameters with the MXene and BlueP/TMDCs hybrid structure phase-sensitive SPR sensor. By the DEPSO, parameters such as composition, thickness and layer number of the SPR sensor could be determined for the best performance. Optimization results obtained by DEPSO were also compared to the conventional layer-by-layer method and the conventional Particle Swarm Optimization (PSO). By these optimized parameters in SPR sensor, the best phase sensitivity could be improved by an order of magnitude when compared to the initial structure. Although objective and result of the work are clear and meaningful, there are several major issues that the authors need to address before it can be recommended for publication. 

Specific points to be addressed:

1. The title of this work is about the DEPSO method for optimizing parameters with the specific SPR sensor structure. More information in Section 3 about the method itself such as how the mutation and crossover step were implemented would be helpful to highlight and support the theme.

2. What are standards for determining initial parameters such as particle number, inertia weight coefficient range and thickness or layer number for different materials in Table 2? Illustration about these conditions would be helpful for better understanding.

3. Does the particle number set in both DEPSO and PSO model affect the final SPR sensitivity result and what is the calculation time for each generation step or the whole simulation workflow? 

4. In Figure 6, it seems that both DEPSO and PSO can optimize the SPR phase sensitivity to a similar level, further discussion of advantages of DEPSO would be helpful. The description on line 266-269 about the relationship between sensitivity enhancement and the optimization method is ambiguous. Does the structure itself enhance the electron concentration when compare to other SPR sensors’ structure or the algorithm can affect the characterization of the material?

5. In the Algorithm:DEPSO part, proper comments or notes could make the algorithm easier to be understood.

6.  It could denote the incident angle at minimal reflectivity in Table 3 and 4 for better comparison.

7. In Figure 5, add legends for colors in different layers and curves in TMM subfigures for better clarity. 

It is good for the most part but some typos and incorrect grammar usages need to be revised. Check case, part of speech and format carefully before resubmitting the script.

Author Response

Comments 1: The title of this work is about the DEPSO method for optimizing parameters with the specific SPR sensor structure. More information in Section 3 about the method itself such as how the mutation and crossover step were implemented would be helpful to highlight and support the theme.

Response 1: Thank you for pointing this out. I agree with this comment. Therefore, I have added the specific steps and equation (8-9) to describe the implementation process of the algorithm in Section 3, line 16-32.

Comments 2: What are standards for determining initial parameters such as particle number, inertia weight coefficient range and thickness or layer number for different materials in Table 2? Illustration about these conditions would be helpful for better understanding.

Response 2: Thank you for pointing this out. At 633 nm, silver film thicknesses (d1, d2) less than 50 nm usually have good sensing performance [16, 18]. And for >10 layers, the resonant dip vanishes. Before invoking the algorithm, the initialization variables are listed in Table 2, all of which are set based on experience that enables the algorithm to perform optimally [40].

Comments 3:  Does the particle number set in both DEPSO and PSO model affect the final SPR sensitivity result and what is the calculation time for each generation step or the whole simulation workflow? 

Response 3: Thank you for pointing this out. In general, the optimization result of an algorithm is related to the number of particles, but it is not the case that more particles means better optimization result of the algorithm. Considering the optimization of this paper, the authors of this paper optimize the objective function in the cases of 100, 200, 300, 400, 500 particles respectively, and the simulation results show that the algorithm can optimize the SPR phase sensitivity to a similar level under these particle numbers. During the optimization process, the angle of incidence is varied in 0.001° steps. Therefore, considering the amount of computation as well as the final optimization time of the algorithm, this paper uses 100 as particle number for optimization. The calculation time for the whole simulation workflow is 511.053s.

Comments 4:  In Figure 6, it seems that both DEPSO and PSO can optimize the SPR phase sensitivity to a similar level, further discussion of advantages of DEPSO would be helpful. The description on line 266-269 about the relationship between sensitivity enhancement and the optimization method is ambiguous. Does the structure itself enhance the electron concentration when compare to other SPR sensors’ structure or the algorithm can affect the characterization of the material?

Response 4: Thank you for pointing this out. In Table 3 and Table 4, I added the iterations of algorithm to further illustrate the advantage of DEPSO. At the same time, in section 4 paragraph 7 which marked in red, I further described the relationship between sensitivity enhancement and the optimization method and the advantages of DEPSO.

‘Compared Figure 5 with Figure 6 and Table 3 with Table 4, it can be seen that the DEPSO algorithm requires fewer iterations to reach optimal solution which shows better convergence characteristics. Besides, DEPSO algorithm exhibits superior global search ability, which enables it to jump out of local optimal value and more fully utilize the high electron concentration and mobility of BlueP/TMDCs and MXene, thus enhancing the performance of SPR sensors, making them suitable for detecting mixed gases. At the same time, with the number of SPR sensor layers increases, the DEPSO algorithm can find the optimal thickness of all layer simultaneously and time saving, which has advantage to work with a large number of design parameters.’

Comments 5:  In the Algorithm: DEPSO part, proper comments or notes could make the algorithm easier to be understood.

Response 5: Thank you for pointing this out. I have added some comments to make the algorithm easier to be understood in the algorithm: DEPSO part.

Comments 6:  It could denote the incident angle at minimal reflectivity in Table 3 and 4 for better comparison.

Response 6: Thank you for pointing this out. I have added the incident angle at minimal reflectivity in Table 3 and 4.

Comments 7:  In Figure 5, add legends for colors in different layers and curves in TMM subfigures for better clarity. 

Response 7: I have adjusted the text information in Figure 5 and added legends for the colors in the curves in the TMM subfigures for better clarity.

Round 2

Reviewer 1 Report

Accept in present form

Reviewer 2 Report

Thank you for revising the paper and replying the comments! The additional information in the revised version provides a more comprehensive perspective to highlight the advantage of DEPSO method.

Your responses to my comments make sense to me. It can be recommended for publication now and please check the writing carefully before finalizing your manuscript.

Please check the writing carefully before finalizing the manuscript.